# Assessments of Muscle Thickness and Tonicity of the Masseter and Sternocleidomastoid Muscles and Maximum Mouth Opening in Patients with Temporomandibular Disorder

**DOI:** 10.3390/healthcare9121640

**Published:** 2021-11-26

**Authors:** Keunhyo Lee, Seungchul Chon

**Affiliations:** Department of Physical Therapy, Graduate School of Medical Science, Konyang University, Daejeon 35365, Korea; lgh7803@naver.com

**Keywords:** masseter muscle, maximum mouth opening, muscle thickness, muscle tone, sternocleidomastoid muscle, temporomandibular disorder

## Abstract

The purpose of this study was to compare the muscle thickness and tone of the masseter and sternocleidomastoid (SCM) muscles and maximum mouth opening (MMO) in subjects with and without temporomandibular disorders (TMD), and perform a correlation comparison on the results of the TMD group. Sixty patients were allocated to the TMD group (*n* = 30) or the non-TMD group (*n* = 30). Ultrasound imaging, myotonometer, and vernier calipers were used to determine the related changes in muscle thickness and muscle tone in masseter and SCM, and MMO, respectively. The TMD group revealed a significant decrease than the non-TMD group in the muscle thickness of masseter and SCM, respectively (*p* < 0.001), with a significant increase in frequency (*p* < 0.001) and stiffness (*p* < 0.001) in the masseter muscle tone, with a significant increase in frequency (*p* < 0.001) and stiffness (*p* = 0.005) in the SCM muscle tone, a significant decrease in the MMO (*p* < 0.001). There was a moderate negative correlation between the relaxed state of masseter muscle thickness and stiffness of SCM muscle tone (*r* = −0.40, *p* = 0.002), and a moderate negative correlation between the relaxed state of SCM muscle thickness and frequency of SCM muscle tone (*r* = −0.42, *p* = 0.001). There was a moderate negative correlation between the clenching state of SCM muscle thickness and the frequency of SCM muscle tone (*r* = −0.47, *p* < 0.001). In addition, a moderate negative correlation between MMO and frequency of SCM muscle tone (*r* = −0.44, *p* < 0.001). The muscle thickness was decreased, and the muscle tone was increased in the masseter and SCM muscle, respectively. Additionally, MMO was decreased in patients with TMD compared with non-TMD.

## 1. Introduction

The temporomandibular joint (TMJ) connects the temporal bone and mandibular bone, and is a complicated joint composed of an articular disc, jaw muscles, and ligaments [1]. It is a joint of multiple functions, such as chewing, deglutition movements, and articulation of the oral cavity [2]. Alessandro et al. [3] reported that the TMJ is a bilateral joint in which the bilateral mandibular bones roll or glide together, so symmetrical movements occur during opening, closing, protrusion, and retrusion. Although it is impossible to move the joints individually, asymmetrical movements occur when moving sideways or when chewing [4]. These asymmetrical movements cause temporomandibular disorder (TMD), causing pain in the soft tissue and limiting functional movement [5].

TMD, which shows pain and functional movement problems, is reported to be experienced by about 75% of the population [6]. In addition to the accompanying pain, malocclusion and crepitation frequently occur while TMD patients open their mouths [7]; in this case, managing activities of daily living are not easy and quality of life may be affected [8]. Additionally, this triggers intraoral pernicious habits, such as clenching and bruxism, thus causing parafunction of the masticatory muscle [9].

The main functions of TMJ, such as mouth opening, chewing, and lateral movement, are controlled by the masticatory, temporal, pterygoid and sternocleidomastoid (SCM), and lateral chewing is considered the most important function [10]. It has been reported that such lateral chewing causes the biomechanical imbalance of masticatory muscles, resulting in TMD [11]. The masseter muscle plays a major role in the chewing function [12], and SCM has been reported as an important muscle providing head and neck stability in performing mastication [13]. Pizolato et al. [14] reported that TMD patients had weak chewing force, and Pereira et al. [15] reported a positive correlation between occlusal force and masseter muscle thickness. 

SCM is an important factor in head control for mastication and is one of the major muscles influencing TMD, referred pain muscle to the stomatognathic systems [16]. Patients with TMD have an imbalanced occlusal pattern [17]. An imbalance in the occlusal pattern promotes an imbalance in SCM activity, causing a lateral tilt of the neck [18]. Pallegama et al. [19] observed a high muscle activity of the SCM in TMD patients. The thickness and activity of the SCM are affected by dysfunction and mandibular movements [17].

The structure of TMJ is stable because the fibrous ligaments protect it from stress and tension in the joint [20]. Abnormalities in the chewing system due to increased masticatory muscle tone affect masticatory dysfunction [21]. Excessive use of the jaw or mouth and constant tension in the head and neck muscles have been reported to experience TMD signs and symptoms [22]. In TMD patients, pain often occurs when the range of motion (ROM) of the jaw joint like mouth open is increased. Restricted mouth opening is a typical symptom of TMD [23]. It is accompanied by symptoms, such as strepitus, limited movement of the jaw joint, and asymmetric mouth movement [24]. 

Emshoff et al. [25] suggested that ultrasonography is a reliable technique for evaluating the masseter and SCM muscles of TMD patients. This study used functional ultrasonographic devices and myotonometers, which are recently universally utilized with high efficiency in the rehabilitation field. The aim of this study was to compare muscle thickness, tone, and maximum mouth opening (MMO) of the masseter and SCM muscles between the TMD group and the normal group using the above quantitative tools. In addition, we aimed to determine is a correlation between each factor like muscle thickness and tone of the masseter and SCM muscles and MMO in the TMD group.

## 2. Materials and Methods

### 2.1. Participants

All subjects were recruited from the Konyang Medical University community. Subjects were classified into TMD and non-TMD through questionnaires based on the American Academy of Orofacial Pain (AAOP). The age ranged from 20 to 27 years, with an average of 24.2 years (Table 1). A physical therapist (K.H.L.) with 5 years of experience made the diagnosis of TMD according to the clinical assessment criteria.

The clinical criteria for diagnosing TMD were determined by a simplified questionnaire based on the AAOP. The questionnaire for TMD is easily identified by relating the patient’s current state to TMD, without a specialized clinical examination [26]. If any of the 10 question factors showed a positive response, it was sufficient to confirm TMD, and the severity of symptoms was determined by the number of positive responses [26] The sensitivity and specificity of this questionnaire were 85% and 80%, respectively [27]. The exclusion criteria were as follows: (1) patients with post-traumatic stress disorder (2) patients with neurological diseases (3) patients who have undergone artificial surgery on the jaw or neck (4) patients who use analgesics and anti-inflammatory drugs. 

Subjects were allocated before the initial assessment. All of them participated in the measurements. The physiotherapist undertaking the assessment was also blinded to the group allocation. All subjects participated after fully understanding the purpose and methods of this study, which provided informed consent. This study was approved by the university ethics and institutional review board (IRB approval no. KYU-2020-169-02 and Clinical Research Information Service approval no. KCT0005941).

### 2.2. Instrumentation

To measure body mass index, a body composition analyzer utilizing a bioelectrical impedance method (In Body 4, Biospace, Seoul, Korea) was used. In this evaluation tool, the intraclass correlation coefficient (ICC) value was body fat percent% (≥0.98), fat mass (≥0.98), and fat free mass (≥0.99) [28]. After removing any metallic items, such as necklaces, earrings, and watches, the subject stood barefoot on a floor electrode and held a handle with each hand, while remaining still.

A numerical rating scale consisting of 11 questions was used to measure the degree of pain within the TMD group. Patients were asked to evaluate their intensity of pain by assigning scores from 0 for “no pain” to 10 for “worst pain imaginable.” In this evaluation tool, the ICC value was 0.95, and the Cronbach alpha value was 0.88 [29].

An ultrasound device was used to measure the thickness of the masseter muscle and SCM in a relaxed and clenched state of the TMJ. This equipment is easy to operate and transport and is accurate for soft tissue evaluation [30]. According to Yamaguchi et al. [31], the intra-rater reliability of ultrasound measurements of the masseter muscle is 0.83 in the rest state and 0.86 in the contraction state in ICC. In addition, ICC was 0.91 in the rest state and 0.92 in the contracted state.

A myotonometer was used to non-invasively measure the tone of the muscles around the TMJ. In addition, it measured the deformation properties of the naturally damped vibrations generated after a short 15 ms mechanical tap on the skin’s surface [32]. The following results were expressed numerically by computerizing the biomechanical properties [33]: (1) the oscillation frequency (Hz), which indicates the tone (i.e., intrinsic tension) of a muscle in a resting state; (2) the logarithmic decrement of a muscle’s natural oscillation, which indicates the elasticity of the muscle; for example, its ability to recover its shape after contraction; (3) dynamic stiffness (N/m), which characterizes the resistance of the muscle to contract. According to a study by Lucy et al. [33], the mean reliability of muscle tone measurement using a myotonometer is very high (ICC > 0.90) of two measurement sets.

A digital vernier caliper was used to evaluate the MMO of the TMJ. This tool can measure in 0.01 mm increments to minimize the error range. It is a versatile precision instrument used to accurately measure point-to-point distances. According to a study by Norman et al. [34], when the MMO was measured using vernier calipers, it was 0.97 in intrasession and 0.97 in intersession.

### 2.3. Procedures

The patients were comfortably measured in a lying position with a pillow under the knees after a stable state for 10 min for ultrasonography and myotonometer measurements. When measuring muscle thickness and muscle tone in the relaxed state, the subject was set so that the head and neck were horizontal, shoulder edges were kept in contact with the floor. Muscle thickness of masseter and SCM muscle were respectively measured in the clenched state that the jaw clenched maximally for 5 s. The measurement was performed on the symptomatic or more painful side as a dominant side. Measured at the end of the relaxed expiration to minimize changes due to breathing. All measurements were conducted three times and the average value was used. 

We used a linear probe (Mysono U6, Samsung Medison Inc., Seoul, Korea) with a broadband frequency of 5–12 MHz. The measurement site was approximately at the thickest part of the masseter muscle in the middle of the mediolateral distance of the ramus [35] and a transversely placed transducer was utilized without applying over-pressure to the skin [36] (Figure 1). According to the method used by Satiroğlu et al. [35], the transducer was held perpendicular to the surface of the skin and special care was taken to avoid excessive pressure during imaging measurements. Scanning the masseter muscle aslant increases the muscle thickness; to avoid this, frequently changed the angle of the transducer until the best echo of the mandibular ramus surface was achieved. The muscle thickness of the SCM was determined using a 7.5 MHz, linear transducer placed approximately 5 cm lateral to the trachea with the neck vertical [37](Figure 2). With the patient in the same position as for the muscle thickness measurements, the muscle belly in the middle of the SCM was palpated and marked.

For the myotonometer measurements (Myoton PRO, MYOTON AS, Tallinn, Estonia), a probe with a diameter of 3 mm was applied perpendicularly to the skin surface with a constant preload of 0.18 N [38]. The tonicity of the masseter and SCM muscle was measured using a myotonometer. A straight line was connected from the subject’s eye corner to the mandibular angle and the intersection point between the straight line and the zygomatic bone was established. The midpoint between the mandibular angle and the intersection point was set as the measuring point of the masseter muscle, which corresponds to the midpoint of the masseter muscle belly [39]. The SCM was examined after palpation at the midpoint between the insertion of the manubrium sterni into the anterior surface and the mastoid process of the temporal bone [40] (Figure 3). The myotonometer was placed with each muscle vertical, and the average of three measurements was used as the data value.

The MMO was measured using a digital vernier caliper (CD-20PSX, Mitutoyo Corp, Kawasaki, Japan), and the subject was seated upright on a fixed chair with armrests, looking straight ahead. The patient was asked to open their mouth to the pain-free range, the distance between the two incisors of the maxilla and mandible was measured.

### 2.4. Statistical Analysis

G-Power 3.1.9.4 software (University of Dusseldorf, Dusseldorf, Germany) was used to perform a sample size calculation. The power (1-β) and alpha levels (α) were set at 0.80 (80%) and 0.05, respectively. In addition, the effect size was set at 0.80. According to prior analysis, the sample size for each group required 26 subjects. In this study, a total of 60 subjects (each group had 30 subjects) was selected for reflecting the drop-out rate before the prior study, but there were no drop-outs. The Kolmogorov–Smirnov test was performed as a normality test method to determine nonparametric/parametric statistics for all measured variables. As a result, it was found to be greater than the significance level of 0.05, confirming the normal distribution. Therefore, the general characteristics of the subjects were expressed as means and standard deviations using descriptive statistics to compare both groups. Independent *t*-tests were used to compare relaxed and clenched muscle thickness, muscle tone, and MMO in both groups. Pearson’s correlation analysis was used to evaluate correlation among muscle thickness, muscle tone, and MMO. The collected data were analyzed using the statistical program SPSS ver. 20.0 (IBM Corp., Armonk, NY, USA). Statistical significance was set at *p* < 0.05.

## 3. Results 

The TMD group revealed a significant decrease compared to the non-TMD group in the muscle thickness of masseter and SCM, respectively, when in the relaxed state (*p* = 0.001) and clenching state (*p* < 0.001) (Table 2). 

The TMD group showed a significant increase compared to the non-TMD group in frequency (*p* < 0.001) and stiffness (*p* < 0.001) in the masseter muscle tone, but decrement (*p* = 0.35) did not show a significant difference. Additionally, the TMD group showed a significant increase compared to the non-TMD group in frequency (*p* < 0.001) and stiffness (*p* = 0.005) in the SCM muscle tone, but decrement (*p* = 0.76) did not show a significant difference (Table 3). 

The TMD group showed a significant decrease compared to the non-TMD group in the MMO (*p* < 0.001) (Table 4). 

There was a mild negative correlation between MMO and frequency of masseter muscle tone (*r* = −0.39, *p* = 0.002). There was a mild negative correlation between MMO and stiffness of masseter muscle tone (*r* = −0.27, *p* = 0.035), with a moderate negative correlation between MMO and stiffness of masseter muscle tone (*r* = −0.51, *p* < 0.001). There was a mild negative correlation between the relaxed state of masseter muscle thickness and frequency of SCM muscle tone (*r* = −0.38, *p* = 0.003), with a moderate negative correlation between the relaxed state of masseter muscle thickness and stiffness of SCM muscle tone (*r* = −0.40, *p* = 0.002). There was a mild negative correlation between the clenching state of masseter muscle thickness and frequency of SCM muscle tone (*r* = −0.34, *p* = 0.008), with a mild negative correlation between clenching state of masseter muscle thickness and stiffness of SCM muscle tone (*r* = −0.32, *p* = 0.011). There was a moderate negative correlation between the relaxed state of SCM muscle thickness and frequency of SCM muscle tone (*r* = −0.42, *p* = 0.001), with a mild negative correlation between the relaxed state of SCM muscle thickness and stiffness of SCM muscle tone (*r* = −0.42, *p* = 0.001). There was a moderate negative correlation between the clenching state of SCM muscle thickness and the frequency of SCM muscle tone (*r* = −0.47, *p* < 0.001) and a mild negative correlation between the clenching state of SCM muscle thickness and stiffness of SCM muscle tone (*r* = −0.31, *p* = 0.015). There was a moderate negative correlation between MMO and frequency of SCM muscle tone (*r* = −0.44, *p* < 0.001), with a mild negative correlation between MMO and stiffness of SCM muscle tone (*r* = −0.34, *p* = 0.007). However, there was no correlation in any other items (Table 5).

## 4. Discussion

The purpose of this study was to determine whether there is a difference in the relaxed and clenched state, muscle thickness, muscle tone, and MMO of the masseter and SCM muscles with and without TMD. As a result, in the TMD group, the masseter and SCM muscle was thinner in the relaxed and clenched states, and also, the length of MMO was found to be short. In addition, in the comparison of muscle tone, both frequency and stiffness of muscles including the masseter and the SCM were shown to be increased in the TMD group. Lastly, in the correlation within the TMD group, the thicknesses of the masseter and SCM muscles decreased as the SCM muscle tone increased. However, the degree of correlation was weak or moderate. 

Widmer et al. [41] found that myopathological damage to the masseter muscle causes not only pain but also the occlusal shape and power of the TMJ and may lead to speech impairment. Therefore, we investigated the thickness of the masseter muscle using quantitative ultrasonographic images of both TMD and non-TMD, respectively. As a result, the thickness of the masseter muscle of the TMD was 18.8% lower in the relaxed state and 15.9% lower in the clenched state. Imanimoghaddam et al. [42] found that the masseter thickness of the TMD group was 10.8% lower than that of a non-TMD. Pereira et al. [15] reported that the masseter muscle thickness of the group with TMD was 3.8% lower in the right contracted state and 2.3% lower in the left contracted state than in the non-TMD group. Castelo et al. [43] found that there is a positive correlation between masseter muscle thickness and bite force. According to Pereira et al. [15], the bite force was significantly lower in the TMD group when compared to the control group. Therefore, it is considered that the TMD group with pain decreases the thickness of the masseter muscle.

In this study, we investigated the muscle thickness of SCM in both TMD and non-TMD groups. As a result, the TMD group showed about 18% thinner SCM muscle in the relaxed and clenched states, respectively. In the Strini et al. [16] study, the thickness of the SCM in the TMD group was 2.8% less than that of the non-TMD group. Strini et al. [17] showed that the thickness of the SCM presented 10.5% less in a relaxed state, and 9.6% less in a clenched state on the right side; the left side was 7.3% less when relaxed and 7.2% less when clenching in the TMD group than in the non-TMD group. The head position and mandibular movements (flexion & lateral tilt) were shown to affect the SCM thickness and activity, especially during the clenching state [17]. In addition, According to Ries et al. [44], the asymmetry activation of SCM muscle was significantly increased in the TMD group when compared to the control group. Therefore, in this study, the thickness of the SCM muscle was decreased in the TMD group.

The measured values of the myotonometer device used in this study were frequency, decrement, and stiffness. Compared with the non-TMD group, the frequency and stiffness of the masseter muscle in the TMD group were 12.6% and 16.5% increased, respectively. In addition, the frequency and stiffness in SCM muscle were increased at 11.5% and 14.4%, separately. Takashima et al. [45] reported that masseter muscle stiffness in the TMD group was approximately 57% increased than that of the non-TMD group. According to Schroeder et al. [46], patients with the TMD often have additional complaints, such as headaches and hypertension in the anterior neck region, bringing about an association of an increase in muscle activity of the SCM. Vain [47] stated that frequency represents muscle tone, decrement means muscle elasticity and the muscle’s ability to restore its initial shape after deformation and stiffness reflects the ability of the muscle to resist changes in its shape. Additionally, it may be considered that the increased muscle tone in the TMD group was affected by external stimuli, such as pain, mal-alignment, overuse [48]. It can be interpreted that both the tone(frequency) of masseter and SCM muscles of the TMD group were increased. 

In this study, the average mouth open range of the TMD group was 3.45 cm, which was 57.1% less than that of the non-TMD group. Rapidis et al. [49] suggest that the normal range of TMJ is 4~5 cm, and if it is less than 3.5 cm it is defined as TMD. Evcik et al. [50] found that MMO in the TMD group was approximately 56.8% less than that of a control group. Therefore, the TMD group was not included in the normal range.

As a result of the correlation study, there were moderate negative results between the MMO and stiffness of masseter muscle tone. There were moderate negative results between the MMO and frequency of SCM muscles tone. Takashima et al. [45] reported that masseter muscle stiffness was negatively correlated with MMO (*r* = −0.389). These kinds of report in correlation with MMO were paralleled with increased muscle tone like stiffness or frequency of masseter and SCM muscle, respectively.

The limitations of this study are as follows. First, the results of this study cannot be generalized because the sample was limited to the young adult population. Second, only a one-sided comparison of the TMJ was performed, not a both-sided comparison. Third, it may be a research design that is somewhat unreasonable to draw conclusions of cause and effect. Therefore, a study investigating the comparison of TMD or non-TMD on a large number of subjects with both sides of TMJ is needed in order to fully elucidate the clinical benefits for a wide range of subjects including temporalis and pterygoid muscles.

## 5. Conclusions

This study provides empirical evidence to show that the thickness of the masseter and SCM muscles decreased in the TMD group when compared to the non-TMD group. In addition, the tone of the masseter and SCM muscles increased, and MMO decreased. Finally, there were moderate negative results between the MMO and stiffness of masseter muscle tone. There were moderate negative results between the MMO and frequency of SCM muscles tone. Therefore, when planning exercise programs and treatments for TMD patients, efforts to reduce the tone of the masseter and SCM muscles are expected to help improve jaw opening.

## Figures and Tables

**Figure 1 healthcare-09-01640-f001:**
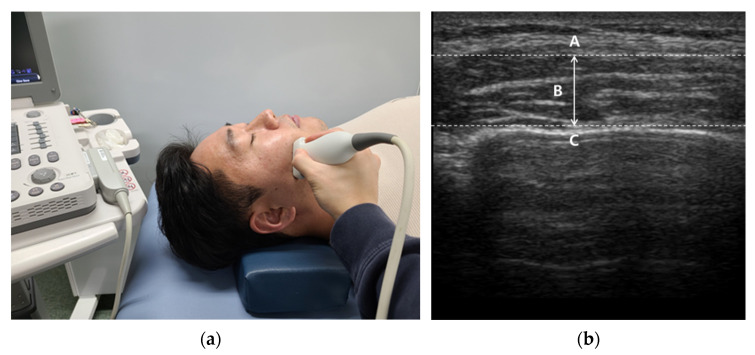
Masseter muscle thickness measurement: (**a**) Measurement site; (**b**) Ultrasound imaging (A: Masseter muscle surface; B: Masseter muscle thickness; C: Mandibular ramus).

**Figure 2 healthcare-09-01640-f002:**
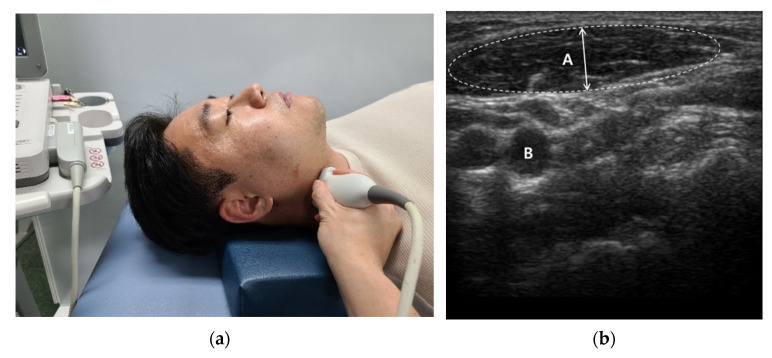
SCM muscle thickness measurement: (**a**) Measurement site; (**b**) Ultrasound imaging (A: SCM muscle thickness; B: Carotid Artery).

**Figure 3 healthcare-09-01640-f003:**
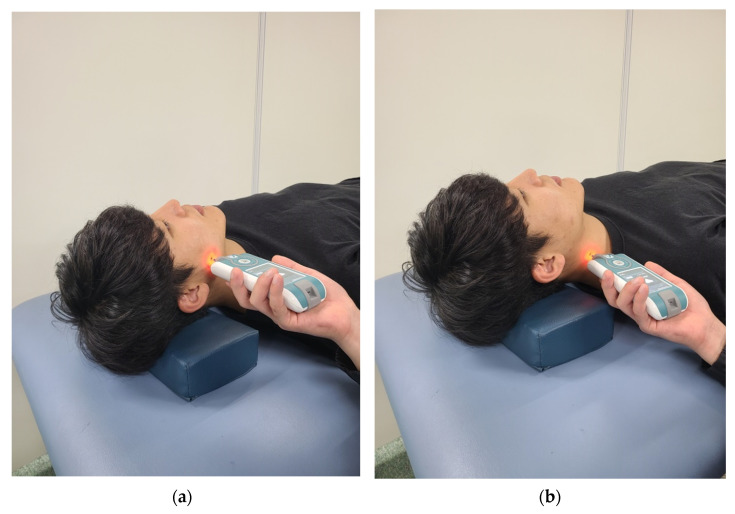
Muscle tone measurement site: (**a**) Masseter muscle; (**b**) SCM Muscle.

**Table 1 healthcare-09-01640-t001:** General characteristics of the subjects (Mean ± SD).

Variable	TMD Group (*n*_1_ = 30)	Non-TMD Group (*n*_2_ = 30)	*p*-Value
Gender (Male/Female)	12/18	14/16	0.60
Age (years)	24.0 ± 1.7	24.3 ± 1.6	0.59
Height (cm)	167.7 ± 5.6	168.3 ± 4.8	0.69
Body mass (kg)	59.9 ± 5.0	60.3 ± 6.9	0.83
BMI (kg/m^2^)	20.9 ± 2.6	21.3 ± 2.6	0.49
NRS	5.3 ± 1.0		
AAOP questionnaire	2.3 ± 1.9		

BMI: Body mass index; NRS: Numerical rating scale; AAOP: American academy of orofacial pain.

**Table 2 healthcare-09-01640-t002:** Comparison on relaxed and clenching thickness in both groups (Mean ± SD).

Variable		TMD Group (*n*_1_ = 30)	Non-TMD Group (*n*_2_ = 30)	*p*-Value
Masseter	Relaxed (mm)	10.7 ± 2.0	12.7 ± 1.9	<0.001
	Clenching (mm)	12.9 ± 2.0	15.0 ± 2.2	<0.001
SCM	Relaxed (mm)	10.0 ± 2.0	11.9 ± 1.6	<0.001
	Clenching (mm)	12.2 ± 2.2	14.4 ± 1.8	<0.001

SCM: Sternocleidomastoid muscle.

**Table 3 healthcare-09-01640-t003:** Comparison on muscle tone in both groups (Mean ± SD).

Variable		TMD Group (*n*_1_ = 30)	Non-TMD Group (*n*_2_ = 30)	*p*-Value
Masseter	Frequency (Hz)	24.1 ± 2.8	21.4 ± 2.4	<0.001
	Decrement (log)	1.5 ± 0.3	1.6 ± 0.3	0.35
	Stiffness (N/m)	507.2 ± 63.4	435.2 ± 71.2	<0.001
SCM	Frequency (Hz)	16.5 ± 1.3	14.8 ± 1.5	<0.001
	Decrement (log)	1.4 ± 0.2	1.4 ± 0.2	0.76
	Stiffness (N/m)	281.6 ± 52.8	246.0 ± 41.2	0.005

SCM: Sternocleidomastoid muscle.

**Table 4 healthcare-09-01640-t004:** Comparison on maximum mouth opening in both groups (Mean ± SD).

Variable	TMD Group (*n*_1_ = 30)	Non-TMD Group (*n*_2_ = 30)	*p*-Value
MMO (cm)	3.4 ± 0.3	5.4 ± 0.5	<0.001

MMO: Maximum mouth opening.

**Table 5 healthcare-09-01640-t005:** Pearson correlations coefficient among muscle thickness, muscle tone and maximum mouth opening in the TMD group.

Variable		Masseter Muscle Tone						SCM Muscle Tone					
		Frequency		Decrement		Stiffness		Frequency		Decrement		Stiffness	
		r	*(p)*	r	*(p)*	r	*(p)*	r	*(p)*	r	*(p)*	r	*(p)*
Masseter	Relaxed	−0.15	(0.25)	−0.03	(0.81)	−0.16	(0.19)	−0.38	(0.003)	−0.04	(0.75)	−0.40	(0.002)
	Clenching	−0.17	(0.17)	−0.01	(0.89)	−0.16	(0.19)	−0.34	(0.008)	−0.02	(0.86)	−0.32	(0.011)
SCM	Relaxed	−0.21	(0.10)	0.02	(0.85)	−0.16	(0.21)	−0.42	(0.001)	0.09	(0.47)	−0.35	(0.005)
	Clenching	−0.21	(0.10)	0.07	(0.56)	−0.27	(0.35)	−0.47	(<0.001)	0.08	(0.53)	−0.31	(0.015)
MMO		−0.39	(0.002)	0.03	(0.77)	−0.51	(<0.001)	−0.44	(<0.001)	−0.07	(0.57)	−0.34	(0.007)

SCM: Sternocleidomastoid muscle; MMO: Maximum mouth opening; 0–0.19 is regarded as very weak, 0.2–0.39 as weak, 0.40–0.59 as moderate, 0.6–0.79 as strong and 0.8–1 as very strong correlation.

## Data Availability

Data from this study can be requested from the first author. The analyzed data is not disclosed to everyone, to protect the privacy and ethical protection of the subjects.

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
