# Peer review of "Assessments of Muscle Thickness and Tonicity of the Masseter and Sternocleidomastoid Muscles and Maximum Mouth Opening in Patients with Temporomandibular Disorder"

_healthcare, 2021, doi:10.3390/healthcare9121640_

Round 1
Reviewer 1 Report
Dear Authors, thank you for submitting your paper.
The aim of the present study was to compare the muscle thickness and tone of the masseter and sternocleidomastoid (SCM) muscles and maximum mouth opening (MMO) in subjects with and without temporomandibular disorders (TMD); and perform a correlation comparison on the results of the TMD group.
I congratulate the authors for this very relevant research, which will add to the dental field. It appears well structured, correctly carried out and written without logical or factual errors
Methodological aspects are deeply cleared in the manuscript.
The topic is in line with the journal aim.
-Data reported in the Methods section are appropriate and precisely described;.
-Results are reported clearly and adequately supported by Tables.
-I suggest to the Authors to improve their reference list citing in the manuscript the following recent articles about TMD in patients affected by juvenile idiopathic arthritis
https://doi.org/10.3390/jcm9041159
https://doi.org/10.3390/children8010033
https://doi.org/10.1186/s40510-021-00380-6
The Conclusions are correctly stated and supported by the findings obtained from the present study.
Author Response
We are grateful for the reviewers’ insightful comments. The manuscript has been extensively revised with adequate clarity to reflect and incorporate the reviewers’ detailed comments throughout the entire manuscript. And also, all changes, including added text were revised by blue color in the corrected manuscript file.
Sincerely yours,
SC Chon and KH Lee
ear Authors, thank you for submitting your paper.
The aim of the present study was to compare the muscle thickness and tone of the masseter and sternocleidomastoid (SCM) muscles and maximum mouth opening (MMO) in subjects with and without temporomandibular disorders (TMD); and perform a correlation comparison on the results of the TMD group.
I congratulate the authors for this very relevant research, which will add to the dental field. It appears well structured, correctly carried out and written without logical or factual errors
Methodological aspects are deeply cleared in the manuscript.
The topic is in line with the journal aim.
-Data reported in the Methods section are appropriate and precisely described;.
-Results are reported clearly and adequately supported by Tables.
-I suggest to the Authors to improve their reference list citing in the manuscript the following recent articles about TMD in patients affected by juvenile idiopathic arthritis
https://doi.org/10.3390/jcm9041159
https://doi.org/10.3390/children8010033
https://doi.org/10.1186/s40510-021-00380-6
The Conclusions are correctly stated and supported by the findings obtained from the present study.
Response: These were corrected as suggested.
(Reference no.5: https://doi.org/10.3390/children8010033)

Reviewer 2 Report
The paper is difficult to understand.
Starting with methods.
The 1st sentence implies that the only criteria is being over age of 20 and reside in a city. The next sentence states that patients were ages 20-27. What type of practice were these patients from, e.g., dental or medical? This is a narrow age range. Was it deliberate? If so, reason? What were the patients coming in for? Exclusion criteria is given in next paragraph. Need to start with WHO is eligible. How they were identified, recruited. The IRB approval and informed consent are noted. Give the information to the reader that I am sure you gave to the IRB.
After stating age range of participants, it is then stated that the patients were randomized to an experimental (?treatment) group and control group. The experimental group being the TMD group and the control group being the non-TMD group. This does not make sense. A person has TMD or they do not. A person cannot be chosen at random and assigned to whether they have TMD. The 2 groups do not appear to have been treated differently in any way, which for a randomized study would be expected. The details read more like a case control study in which the physiotherapist was blinded as to case status.
The methods used in measurements are described very well. Statistics are appropriate.
The results are also fairly well presented.
Presentation suggestions:
- As this is a small study, in presentation of means and standard deviations/errors, only one decimal place at most should be used. This paper, by nature of topic and design, is number “heavy”. Where possible need to decrease the amount of numbers present, and to make what you present as clear as possible.
- for r’s, use only 2 decimal places,
- p-values of ‘.000’ would be better presented as <.001.
- for other p-values, specifically those >0.1, consider using at most 2 decimal places. In reality, only 1 decimal place is needed to inform the reader of the lack of statistical significance (and is my preference). The 2nd and especially the 3rd decimal place relates no valid information.
- In Table 5, align better. Reducing the number of decimal places will help. Considering putting parentheses around p-values to more easily distinguish from r’s.
The discussion is adequate. Usually the strengths of the study are presented succinctly after the limitations.
Author Response
We are grateful for the reviewers’ insightful comments. The manuscript has been extensively revised with adequate clarity to reflect and incorporate the reviewers’ detailed comments throughout the entire manuscript. And also, all changes, including added text were revised by blue color in the corrected manuscript file.
Sincerely yours,
SC Chon and KH Lee
The paper is difficult to understand.
Starting with methods.
The 1st sentence implies that the only criteria is being over age of 20 and reside in a city. The next sentence states that patients were ages 20-27. What type of practice were these patients from, e.g., dental or medical? This is a narrow age range. Was it deliberate? If so, reason? What were the patients coming in for? Exclusion criteria is given in next paragraph. Need to start with WHO is eligible. How they were identified, recruited. The IRB approval and informed consent are noted. Give the information to the reader that I am sure you gave to the IRB.
Response: These were corrected as suggested
After stating age range of participants, it is then stated that the patients were randomized to an experimental (?treatment) group and control group. The experimental group being the TMD group and the control group being the non-TMD group. This does not make sense. A person has TMD or they do not. A person cannot be chosen at random and assigned to whether they have TMD. The 2 groups do not appear to have been treated differently in any way, which for a randomized study would be expected. The details read more like a case control study in which the physiotherapist was blinded as to case status.
Response: This randomization was deleted as suggested
Presentation suggestions:
As this is a small study, in presentation of means and standard deviations/errors, only one decimal place at most should be used. This paper, by nature of topic and design, is number “heavy”. Where possible need to decrease the amount of numbers present, and to make what you present as clear as possible.
for r’s, use only 2 decimal places, p-values of ‘.000’ would be better presented as <.001.
for other p-values, specifically those >0.1, consider using at most 2 decimal places. In reality, only 1 decimal place is needed to inform the reader of the lack of statistical significance (and is my preference). The 2nd and especially the 3rd decimal place relates no valid information.
In Table 5, align better. Reducing the number of decimal places will help. Considering putting parentheses around p-values to more easily distinguish from r’s.
Response: These were corrected as suggested (Table 1-5).

Reviewer 3 Report
Dear authors,
The manuscript is overall interesting. However, there are several questions to be addressed prior to acceptance. All comments are described in the attached file.
My best wishes.

Author Response
We are grateful for the reviewers’ insightful comments. The manuscript has been extensively revised with adequate clarity to reflect and incorporate the reviewers’ detailed comments throughout the entire manuscript. And also, all changes, including added text were revised by blue color in the corrected manuscript file.
Sincerely yours,
SC Chon and KH Lee
Introduction:
1) (page 1) The temporomandibular joint (TMJ) connects the temporal bone and mandibular
bone, and is a complicated joint composed of an articular disc, jaw muscles, and ligaments. / Reference
Response: These were corrected as suggested
2) (page 1) Although it is impossible to move the joints individually, it is said that this is asymmetrical during lateral movements and chewing. / Reference. Please, do not use terms such as “it is said”. Be objective in your statements based on references.
Response: These were corrected as suggested
3) (page 2) we determined if there / “we aimed to determine”. Please, restructure
Response: These were corrected as suggested
4) (page 2) We believe the results of this study can be used as clinical data for TMD treatment programs in physical therapy part in the future. / How? The introduction must provide a better rationale to support that statement.
Response: These were deleted as suggested
Material and Methods:
1) (page 2) This study consisted of sixty adults / Describe how the participants were recruited.
Response: These were deleted as suggested
2) (page 2) experimental group / Do not use “experimental” as the study design is not experimental, but observational(cross-sectional). I suggest the term TMD and non-TMD groups.
Response: These were corrected as suggested
3) (page 2) Randomization was done with sealed envelopes. The sealed letters for the experimental group (TMD group) and the control group (non-TMD group) were arranged by the investigator. / Please, explain the reason for randomization. In my view point, the diagnosis of TMD would support the allocation.
Response: This randomization was deleted as suggested
4) (page 3) The criteria for diagnosing TMD were determined by a simplified questionnaire based on the American Academy of Orofacial Pain. The questionnaire for TMD easily identifies by relating the patient’s current state to TMD, without a specialized clinical examination [24]. If any of the 10 question factors showed a positive response, it was sufficient to confirm TMD, and the severity of symptoms was determined by the number of positive responses [24]. / Please, provide the validity, reliability and/or responsiveness of this tool. Was this questionnaire tested?
Response: sensitivity and specificity of the questionnaire was newly added
5) (page 3) Table 1. General characteristics of the subjects (Mean ± SD). / Is there a score for the questionnaire? If so, please insert in table 1.
Response: AAOP questionnaire was newly added
6) (page 3) Table 1. Gender (Male/Female) / Please, provide a chi-square test to ensure the same frequency of male/female.
Response: These were corrected as suggested
7) (page 3) Table 1. NRS / Was the control group asked about pain? If so, please insert the values and the comparison (p-value).
Response: The non-TMD group was not asked about pain.
8) (page 3) We used a linear probe with a broadband frequency of 5-12㎒ / Please, add the equipment detailed description, including the manufacturer, country, model, etc.
Response: These were corrected as suggested
9) (page 5) myotonometer / Please, add the equipment detailed description, including the manufacturer, country, model, etc.
Response: These were corrected as suggested
10) (page 5) digital vernier caliper / Please, add the equipment detailed description, including the manufacturer, country, model, etc.
Response: These were corrected as suggested
11) (page 5) 2.3. Instrumentation / I suggest to describe the instruments previously to procedures.
Response: These were corrected as suggested
12) (page 5) a body composition analyzer utilizing a bioelectrical impedance method (In Body 4, Biospace, Seoul, Korea) was used. / Please, provide equipment's validity, reliability and/or responsiveness data.
Response: intraclass correlation coefficient was newly added
13) (page 6) 2.4. Statistical analysis / Please consider an effect size analysis. This would be helpful for clinical purposes. The effect size was set at 0.80 / Was this value extracted from previous studies? Please, clarify.
Response: Yes, this effect size was extracted from pilot study. This was corrected as your suggestion.
Discussion:
1) (page 8) Therefore, it is considered that TMD with pain decreases the bite force, indicating a decrease in thickness of masseter muscle. / The bite force was not assessed. Please, limit your conclusions to your results. Also, be careful in your statements. The cross-sectional design does not allow any cause-effect inferences.
Response: These were corrected as suggested
2) (page 8) Therefore, in this study, the dysfunction of the TMD is considered to decrease the thickness of the SCM muscle. / Be careful with your statements. This is a cross-sectional design, not allowing any cause-effect inferences. Please, restructure.
Response: These were corrected as suggested
3) (page 9) frequency. / Frequency of what? Please, clarify.
Response: These were corrected as suggested
4) (page 9) Therefore, it is considered that the TMD group may be accompanied by functional difficulties because it is not included in the normal range. / Again, you did not assess funcionality. Leave your comments restricted to your results.
Response: These were corrected as suggested
5) (page 9) The limitations of this study are as follows. / Include the study's design as a limitation to conclude about cause-effect.
Response: “Third, it may be a research design that is somewhat unreasonable to draw conclusions of cause and effect” This was newly added as your suggestion.
6) (page 9) small sample size / It is small. But you performed a sample size calculation. So, consider this in this section.
Response: These were revised(deleted) as suggested.
Conclusions:
1) (page 9) TMD group decreased the thickness of the masseter and SCM muscles / TMD group had decreased... compared to non-TMD group. Your conclusions must follow the objetives and be restricted to what the study design allows..
Response: These were corrected as suggested
Data Availability Statement: Data from this study can be requested from the first author. The analyzed data is not disclosed to everyone, to protect the privacy and ethical protection of the subjects. / The privacy is protected by blinding the reader for names and any possibility to identify the participant. Now, transparency is essential to ensure the credibility of your study. Consider depositing the raw data in a repository system, such as Mendeley.
Response: Yes, I agree with you. I’ll upload my raw data.

Round 2
Reviewer 2 Report
See attached document. My original comments were not adequately addressed.

Author Response
We are grateful for your insightful comment. Again, the manuscript was revised with adequate clarity to reflect your detailed comment. And also, all changes, including added text were revised by blue color in the corrected manuscript file.
Sincerely yours,
SC Chon and KH Lee
Reviewer response: No, they were not corrected! You now state “ … was conducted with 60 adults (residents of metropolitan cities) who voluntarily consented through the application documents. …” This does NOT state who was eligible. Was being 20-29 years of age an eligibility criteria? How were they identified, e.g., where and how did they access the application documents. Was this online, in a doctor’s office, dentist office, rehab facility? WHERE were they identified/recruited from? You are not telling the reader anything. What did you put in the IRB Protocol?
Response: These were corrected as suggested
“This study was conducted with 60 adults (residents of metropolitan cities) who voluntarily consented through the application documents” → “Recruitment of subjects for this study was posted on the bulletin board of K-University, and subjects were conducted for those who are interested or concerned about the temporomandibular joint disorder”
Reviewer response: The word randomized was deleted, as were the terms “treatment” and “control”. However, you still “assigned” participants to a TMD or non-TMD group. This does not make sense! A person either has this condition or they do not – you cannot assign someone as having the condition or not. It reads as though you have enrolled 60 people and then arbitrarily said whether or not they had TMD. Consider the following statement: 60 people were enrolled, 30 were assigned to have cancer and 30 as not having cancer. It does not make sense! Was the “diagnostic questionnaire” administered to the non-TMD group? If it was, and it should have been, the score should have been zero. I am guessing that somewhere, online or in an office, a person aged 20-29 could express interest in the study and the 1st 30 with and the 1st 30 without TMD were enrolled. You need to tell the reader!
Response: These were corrected as suggested
“and the subjects are as follows: 30 patients were assigned to an TMD group and 30 normal adults to a non-TMD group. The subjects’ ages ranged from 20 to 27 years, and the average was 24.2 years (Table 1).” → “Subjects of this study were classified into TMD and non-TMD through questionnaires based on the American Academy of Orofacial Pain (AAOP). The number of participants in both groups was 30 respectively, and the age ranged from 20 to 27 years, with an average of 24.2 years (Table 1).”
Reviewer response: The authors fully addressed the suggestions in Tables 1-4, and partly in Table 5. Should reduce the number of decimal places for r’s to two in Table 5 and in the abstract.
Response: These were corrected as suggested

Reviewer 3 Report
Dear authors,
Congratulations for all corrections in your manuscript.
However, I need to raise the Effect Size issue again. Please, consider the ES analysis in all pairwise comparisons establishing the level of relevance of your findings.
Regards.
Author Response
We are grateful for your insightful comment. Again, the manuscript was revised with adequate clarity to reflect your detailed comment. And also, all changes, including added text were revised by blue color in the corrected manuscript file.
Sincerely yours,
Warm regards
SC Chon and KH Lee
Dear authors,
Congratulations for all corrections in your manuscript.
However, I need to raise the Effect Size issue again. Please, consider the ES analysis in all pairwise comparisons establishing the level of relevance of your findings.
Response: These were corrected as suggested.

Round 3
Reviewer 2 Report
The authors have now appropriately described how the study sample was identified and classified.
I have one very minor point: in Table 5 parentheses should also be around the (p) column heading as it is around the p-values in the table.